# Quantitative and Qualitative Assessment of European Catfish (*Silurus glanis*) Flesh

Cristina Simeanu [1], Emanuel Măgdici [1], Benone Păsărin [1], Bogdan-Vlad Avarvarei [1,*] and Daniel Simeanu [2]

1 Department of Animal Resources and Technology, Faculty of Food and Animal Sciences, "Ion Ionescu de la Brad" University of Life Sciences, 8 Mihail Sadoveanu Alley, 700489 Iasi, Romania
2 Department of Control, Expertise and Services, Faculty of Food and Animal Sciences, "Ion Ionescu de la Brad" University of Life Sciences, 8 Mihail Sadoveanu Alley, 700489 Iasi, Romania
* Correspondence: bvavarvarei@uaiasi.ro

**Abstract:** Quantitative and qualitative flesh production in the *Silurus glanis* species was comparatively studied between two fish groups: one from aquaculture (AG) and the other from a natural environment, the Prut River (RG). Morphometry was carried out on the fish, and then biometric and conformational indices were calculated. Better values were found in the aquaculture catfish. The Fulton coefficient was 0.82 in the Prut River fish and 0.91% in the farmed ones. The fleshy index reached 19.58% in the AG fish and 20.79% in the RG fish, suggesting better productive capabilities in the AG fish. Postslaughter, the flesh yield and its quality were assessed at different moments throughout the refrigeration period (0–15 days), and chemical compound loss occurred. In the AG samples, the water content decreased by 8.87%, proteins by 27.66%, and lipids by 29.58%. For the RG samples, the loss reached 8.59% in water, 25.16% in proteins, and 29%in lipids. By studying the fatty acids profile and sanogenic indices, good levels of PUFA (31–35%) were found, and the atherogenic index reached 0.35–0.41 while the thrombogenic index ranged between 0.22 and 0.27. Consequently, it can be stated that fish origin and especially the refrigeration period influence the flesh proximate composition and nutritional value of European catfish.

**Keywords:** European catfish; somatometry; corporal indice; flesh yield; nutritional quality

## 1. Introduction

The *Silurus glanis* L., 1758 species is the main European representative of the *Siluriformes* order. They are predatory fishes with quite aggressive behaviour, hunting during the night or even in daytime when waters are murky, relying mainly on nonvisual sensitive organs [1,2]. They easily adapt to environmental conditions and are mostly found in freshwaters from Central and Eastern Europe and Southwest Asia but occasionally can be found in the salty waters of the Black Sea and Baltic Sea [3,4]. They are a large and very aggressive fish and have reached certain lakes and rivers in Western Europe where they developed quite explosively and sometimes detrimentally to the local fish species [5,6]. European catfish farming has been an increasing trend throughout the last years due to three main factors: it has a tasteful flesh, particularly appreciated by consumers, especially in Eastern Europe and Asia [3,7,8]; it is a useful species in fisheries practising polyculture, providing a good health state of biocenosis [9,10]; and last but not least, it is very appreciated in sport fishing in countries like Spain, the Netherlands, France, and Italy [11–13] due to its impressive dimensions and the struggle during the drill. Catfish rearing in different aquaculture systems is closely correlated to the natural environment. According to the FAO statistics on aquaculture and fisheries, the catfish spreading and rearing area in Europe covers 19 countries (Belarus, Bulgaria, Croatia, Czech Republic, France, Germany, Greece, Hungary, Lithuania, Moldova Rep., North Macedonia, Poland, Romania, Russian Federation, Serbia, Slovakia, Slovenia, Switzerland, and Ukraine), and the reported production was of 7554 tons in 2019 [14]. The second continent in terms of yield of *Silurus glanis* is Asia,

which is mostly spread across seven countries (Azerbaijan, Iran, Kazakhstan, Tajikistan, Turkey, Turkmenistan, and Uzbekistan), and the reported production in 2019 was 3851 tons [14].

In Romania, catfish farming is less developed in comparison with cypriniculture and salmoniculture. From 2010–2019, catfish flesh yield in Romania reached 1826 tons (165 tons in 2019), ranking Romania in tenth place worldwide. A very important role in achieving such production belongs to the Danube River and Danube Delta, from where, according to the NAFA (National Agency for Fishery and Aquaculture), the largest quantity is harvested [14]. Catfish have morphological, anatomical, and physiological adaptations that allow them to occupy other habitats different from flowing waters, such as natural and artificial lakes, favouring their farming in monoculture or polyculture systems along with carp or other cyprinids. In this way, fish farms represent the second provenance source of European catfish flesh in Romania.

European catfish is one of the species leaving strong impressions in consumers' sensory memory, and consumers have benefitted from an increasing trend among farmers throughout the last years. The flesh is highly appreciated by consumers due to its average content of lipids and high content of proteins but also due to its remarkable sensory traits, especially developed after some special cooking processes [15,16].

In the current paper, we aimed to:

- Obtain quantitative data by gravimetry and morphometry run for certain anatomical parts of European catfish originating from aquaculture and a natural environment;
- Calculate biometric indices to assess fish productive potential, maintenance state, and adaptability to provided environment conditions;
- Track the dynamics of flesh chemical composition, fatty acids profile, and sanogenic indices under the influence of the lastingness of refrigeration period (up to 15 days).

The practical utility of the study is given by the knowledge and understanding of the qualitative and quantitative characteristics of *Silurus glanis* flesh issued from different rearing environments and preserved by refrigeration, so we could indicate which meat is of better quality in terms of origin and shelf life.

## 2. Materials and Methods

### 2.1. Biological Studied Material

Three hundred individuals of *Silurus glanis* of various sizes and weights issued from two different rearing environments represented by "Acvares" fish farm situated in Iași County, Romania (aquaculture, geo coordinates (47°19′23.4″ N–27°31′08.8″ E)) and by Prut River on sector Bivolari–Gorban, Iași County, Romania (geo coordinates: 47°32′01″ N 27°26′29″ E–46°53′43″ N 28°05′07″ E) (capture) were studied. Biological material from aquaculture was gathered using a trawl and was stored alive in a submersed storage cage in a waiting pool.

Fish from Prut River were captured by using sport fishing equipment composed of a rod and reel. The sector of Prut River where fish were captured is the eastern boundary of Iasi County, Romania.

The groups were coded in relation to fish origin:

➢ AG—aquaculture group, individuals/samples from farmed fish;
➢ RG—river group, individuals/samples from Prut River (capture).

Body mass of the fish ranged between 1300 g and 2200 g. This weight range for studied specimens is specified by the literature (better development rhythm combined with economics of production) [17–22] and the market demand (buyers' preference on average fish weight).

### 2.2. Physical–Chemical Parameters of Water

Water temperature (°C) had close values for both rearing systems. In the studied period (March-October), in March, water reached a temperature of 10.2 °C in system AG

and 9.2 °C in system RG. The highest thermal values were recorded in July (25.5 °C) for fish farms, respectively, in August (24.6 °C) for Prut River. At the beginning of the studied period (March–August), a slight difference in temperature was observed, with higher values in farm starting in September; these were slightly higher in Prut River. Water pH varied between 7.2 and 7.9 on farms while in Prut River water, it ranged between 7.1 and 7.5, considered normal for good development of studied species [23]. Quantity of dissolved oxygen ranged between 4.09 mg/L and 8.85 mg/L in farm water while in natural environment, dissolved oxygen varied from 8.06 mg/L to 10.12 mg/L, values considered normal for the regular development of catfish. The other physical–chemical parameters of the water analysed in the current study were chlorides ($Cl^-$, 60.21–105.91 mg/L in farm and 74.26–101.34 mg/L in Prut River); nitrites, $NO_2$, 0.02–0.15 mg N/L in farm and 0.08–0.21 mg N/L in Prut River; nitrates, $NO_3$, with values from 0 up to 2.51 mg N/L in farm and between 1.12–2.14 mg N/L in Prut River; ammonium ($NH_4^+$) with values from 0.03 till 0.14 mg N/L in farm and between 0.14–0.35 mg N/L in Prut River; and phosphates ($PO_4^3$) with values from 0 till 0.12 mg P/L in fish farm and between 0.10–0.21 mg P/L in Prut River. The water from both environments was suitable for a normal development of fish, placed in 2nd and 3rd quality categories specific for fish farming systems [23].

Rearing of European catfish on farm was realised in 2 ponds of 30 hectares each in polyculture with carp (*Ciprinus carpio*) (82%) and silver carp (*Hypophthalmichtys molitrix*) and bighead carp (*Aristichtys nobilis*) (12%) aged one year older than the European catfish with body masses of around 500 g. Rate of European catfish in both ponds was 6% (750 individuals). The catfish had masses of around 250 g at brooding moment and between 1.3 kg and 2.2 kg at the end of growth.

### 2.3. Catfish Feeding

Feeding of catfishes from aquaculture was realised with mixed feed, providing between 120–180 kg/day (monthly variable) in 6 portions of 20–30 kg. Mixed feed was purchased from the specialised market with the following characteristics: granule dimension–6 mm, dry matter 89%, crude protein 54%, crude fat 20%, crude ash 9%, crude fibre 1%, P 1.1%, vitamin A 15000 IU/kg, vitamin D3 1800 IU/kg, vitamin E 105 mg/kg, and vitamin C 280 mg/kg. Energetic value was 20.6 MJ/kg digestible energy.

Catfishes from Prut River benefited from feed sources that naturally occur in the river, consisting of earthworms, snails, insects, tadpoles, frogs, and fish, such as carp (*Ciprinus carpio*), Gibel carp (*Carassius auratus gibelio*), common bream (*Abramis brama*), common bleak (*Alburnus alburnus*), chub (*Leuciscus cephalus*), common rudd (*Scardinius erythrophthalmus*), and common dace (*Leuciscus leuciscus*).

### 2.4. Morphometric and Gravimetric Assessments

Morphometric assessments were based on the literature methodology [24–32], and the measurements were run using an ihtyometer. Other measuring instruments used in determination of metric characters were graded ruler, tape measure, square, callipers, and tape line.

Growth performances and the main corporal indices were assessed via morphometry based on the anatomical keypoints highlighted in Figure 1 [25,26].

Gravimetric assessments were run using a PGW 6002 I precision scale and a Partner AS220/C/2 analytical scale.

Dressed yield after slaughter was calculated using Equation (1) [20,21]:

DY (%) = (carcass mass or anatomical analysed portion × 100)/initial mass of live fish     (1)

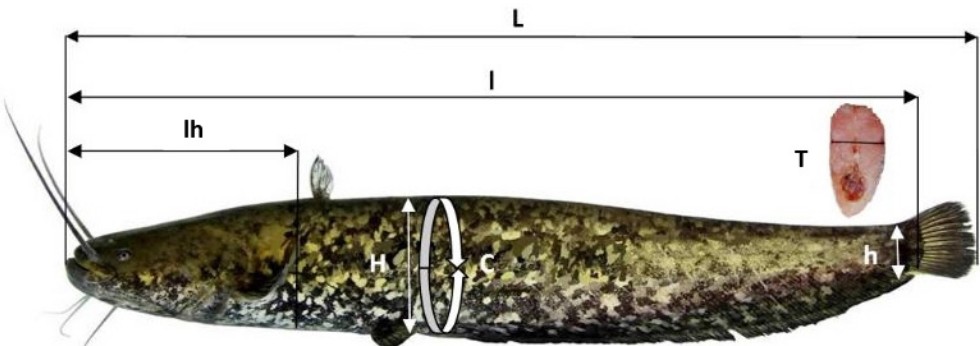

**Figure 1.** Body morphometry in European catfish (original photo): L—total length of fish; l—standard length of body; lh—length of head; H—maximal height of body; h—minimal height of body; C—maximal circumference of body; T—maximal thickness of body.

### 2.5. Body Indices and Coefficients

For reaching the aims of the current research, certain corporal indices were calculated:

Profile index (height) points out the corporal format of studied individuals facilitating their framing into a certain profile [20,21] in accordance with Equation (2):

$$PI = l/H, \tag{2}$$

where PI = profile index; l = body standard length (cm); and H = body maximal height (cm).

Fulton coefficient (maintenance index) indicates a direct proportionality ratio between its values and maintenance state of studied individuals and was calculated using Equation (3) [33]:

$$FC = (m * 100)/l^3, \tag{3}$$

where FC (Fulton coefficient) (maintenance index) (%); m = corporal mass (g); and l = fish standard length (cm).

Quality index (Kiselev) is based on Kiselev relation offering data regarding quality of fishery material. This index was calculated in accordance with Equation (4):

$$QI = l/C, \tag{4}$$

where QI = quality index; l = body standard length (cm); and C = body maximum circumference (cm).

Thickness index represents the existing ratio between body maximal height and its maximal thickness in accordance with Equation (5):

$$TI = (T/H) * 100, \tag{5}$$

where TI = thickness index (%); T = body maximal thickness (cm); and H = body maximal height (cm).

Fleshy index was calculated in accordance with Nistor et al. (2012) [34,35] and expresses the percentage rate of head from body standard length using Equation (6):

$$FI = (lh/l) * 100, \tag{6}$$

where FI = fleshy index (%); lh = head length; and l = body standard length (cm).

The equations used in body indices and coefficients computation were provided by the literature [36–38].

### 2.6. Sampling

Consecutive to morphometry, fish was preserved via refrigeration in sealed plastic recipients and then trenched and filleted to obtain skinless fillets. Samples had a mass of

100 g and were individually refrigerated at temperatures between 2 °C and 4 °C and an air relative moisture of 80–85%.

### 2.7. Assessment of Flesh Chemical Composition

Flesh water content was assessed via oven drying method, so the analysed sample was subjected to drying at 105 °C temperature as specified by Romanian Standard SR ISO 1442/1997.

Assessment of proteins was realised in accordance with AOAC official methods of analysis/1990 [39–42], compatible with Romanian Standard SR ISO 937:2007, by using a Velp Scientifica device (DK6 digestion unit and UDK7 distillation unit) following the Kjeldhal method.

Soxhlet method was applied to measure fat content on a Velp Scientific–SER 148 device following the manufacturer specifications as well as the AOAC official methods of analysis/1990 [39–42], compatible with Romanian Standard SR ISO 1443:2008 [39,43].

To determine the crude ash (total minerals) content, the sample was subjected to calcination method in an electric muffle furnace at a working temperature of +550 °C, in accordance with the Romanian Standard SR ISO 936:1998.

### 2.8. Analysis of Fatty Acids Profile and Nutritional Quality of Lipids

The extraction and quantification of FAME (fatty acid methyl esters) from European catfish flesh was realised by detection through gas chromatography and mass spectrometry on a Perkin Elmer chromatographic device connected with a mass spectrometer detector (GC-MS) [44–47].

Fatty acids were quantified as g FAME/100 g of total identified FAME [44–47]. For lipid profile, fatty acids were grouped as follows:

Saturated fatty acids (SFA) as $\Sigma$ SFA = C10:0 + C12:0 + C14:0 + C15:0 + C16:0 + C18:0 + C20:0 + C22:0;

Monounsaturated fatty acids (MUFA) as $\Sigma$ MUFA = C16:1 + C18:1 *cis*–9 + C20:1 n−9 + C22:1 n−9;

Polyunsaturated fatty acids (PUFA) as $\Sigma$ PUFA = C18:2 n−6 + C20:2 n−6 + C18:3 n−3 + C20:3 n−3 + C20:5 n−3 + C18:3 n−6,

Total unsaturated fatty acids as sum of MUFA and PUFA [45,46,48].

The quantities of $\Omega$-3 and $\Omega$-6 PUFA series were expressed as a rate (n−3/n−6).

Polyunsaturation index (PI) of European catfish flesh was calculated in accordance with Equation (7), established by Timmons [45,48,49]:

$$PI = \text{C18:2 n−6} + (\text{C18:3 n−3} \times 2). \tag{7}$$

AI (atherogenic index) and TI (thrombogenic index) of fats were run on the basis of FAME GC analysis for European catfish flesh in accordance with Equations (8) and (9), established by *Ulbricht and Southgate* [44,48,50,51]:

$$AI = (\text{C12:0} + \text{C16:0} + 4 \times \text{C14:0})/[\Sigma \text{ MUFA} + \Sigma \text{ (n-6)} + \Sigma \text{ (n-3)}], \tag{8}$$

$$TI = (\text{C14:0} + \text{C16:0} + \text{C18:0}) / [0.5 \times \Sigma \text{ MUFA} + 0.5 \times \Sigma \text{ (n-6)} + 3 \times \Sigma \text{ (n-3)} + \Sigma\text{(n-3)} / \Sigma\text{(n-6)}]. \tag{9}$$

Equation (10), published by Fernandez et al. [51], was utilised in calculation of rate between fatty acids with hypocholesterolemic (h) and hypercholesterolemic (H) effects.

$$\text{h/H (hypocholesterolemic / Hypercholesterolemic)} = (\text{C18:1} + \text{PUFA})/(\text{C12:0} + \text{C14:0} + \text{C16:0}) \tag{10}$$

### 2.9. Data Analysis

The main experimental data (50 repetitions per biometric traits, body indices, and yields of cut parts/group and 6 analytical repetitions for analytical chemistry investigations/group) were statistically processed to obtain the arithmetic mean and standard error

of mean. Statistical significance of differences between samples was investigated via Fisher testing [52] included within the Data Analysis ToolPack—ANOVA single factor, Microsoft Excel 2019 software.

## 3. Results

### 3.1. Morphometry and Body Indices

The measurements of certain anatomical portions in the case of European catfish offer only strict quantitative information. Calculations of certain rates between dimensions also provide qualitative information regarding a productive potential [53,54].

The main biometric investigations are presented in Table 1. An average live weight of 1840.71 g was measured in the AG group, which is 3.12% higher than the RG one. Moreover, a 7.89% higher maximum body height (10.80 cm) was found in the AG fish compared to the RG fish. The average total length of the AG fish was 63.45 cm, 8.07% lower than in the RG fish. The values for the AG fish were lower by 2.16% for standard length, 7.77% for head length, 2.76% for maximum circumference, and 1.97% for maximum body thickness compared to the RG fish. The data in Table 1 highlight a better body development of aquaculture fish vs. river environment fish.

**Table 1.** Main biometric assessments in European catfish (*Silurus glanis*).

| Biometric Traits | RG (n = 50) | | | AG (n = 50) | | | *p*-Value |
|---|---|---|---|---|---|---|---|
| | Mean ± SEM | Min. | Max. | Mean ± SEM | Min. | Max. | |
| Body mass (g) | 1784.91 ± 37.43 | 1252.6 | 2192.5 | 1840.71 ± 30.25 | 1386.0 | 2152.4 | 0.0793 |
| Total length (cm) | 68.51 ± 0.61 | 58.53 | 75.12 | 63.45 ± 0.42 | 58.9 | 65.7 | 0.0041 |
| Standard length (cm) | 60.04 ± 0.40 | 54.3 | 63.7 | 58.74 ± 0.46 | 51.4 | 63.6 | 0.1059 |
| Head length (cm) | 12.48 ± 0.12 | 10.62 | 13.39 | 11.50 ± 0.11 | 9.6 | 12.4 | 0.0028 |
| Body maximum height (cm) | 10.01 ± 0.11 | 8.4 | 11.3 | 10.80 ± 0.10 | 9.5 | 11.8 | 0.0075 |
| Body maximum circumference (cm) | 32.18 ± 0.37 | 26.0 | 37.3 | 31.29 ± 0.21 | 28.6 | 32.9 | 0.0964 |
| Body maximum thickness (cm) | 7.08 ± 0.09 | 5.9 | 8.0 | 6.94 ± 0.06 | 6.1 | 7.8 | 0.1868 |

SEM: Standard error of mean. Analysis of variance: check *p* values per row.

The main body indices in the studied fish populations are comparatively presented in relation to their origin (RG—river; AG—aquaculture) in Table 2.

**Table 2.** Main body indices in *Silurus glanis* related to fish origin.

| Calculated Index | RG (n = 50) | | | AG (n = 50) | | | *p*-Value |
|---|---|---|---|---|---|---|---|
| | Mean ± SEM | Min. | Max. | Mean ± SEM | Min. | Max. | |
| Profile index | 6.00 ± 0.04 | 5.47 | 6.58 | 5.44 ± 0.03 | 5.00 | 5.91 | 0.0002 |
| Fulton coefficient | 0.82 ± 0.01 | 0.74 | 1.03 | 0.91 ± 0.01 | 0.78 | 1.10 | 0.0063 |
| Quality index | 1.87 ± 0.02 | 1.57 | 2.03 | 1.88 ± 0.01 | 1.77 | 2.01 | 0.0947 |
| Thickness index | 70.77 ± 1.04 | 59.02 | 84.28 | 64.29 ± 0.61 | 57.65 | 70.98 | 0.0029 |
| Fleshy index | 20.79 ± 0.19 | 18.18 | 23.19 | 19.58 ± 0.14 | 17.77 | 21.53 | 0.0082 |

SEM: Standard error of mean. Analysis of variance: check *p* values per row.

Calculation of the profile index for the AG fish revealed an average of 5.44, which is 9.33% lower than the RG fish, underlining a better development of dorsal muscle mass in the aquaculture fish. The Fulton coefficient presented 10.97% better values in the AG fish; moreover, the quality index had slightly higher values by 0.53% in the AG vs. RG group, suggesting as well a better development of the aquaculture catfish than the ones captured from the natural environment. The thickness index reached 70.77 in the RG fish, which is 10.07% higher than the AG fish, suggesting that river catfish are more robust and better adapted to the natural environment.

*3.2. Quantitative Flesh Production*

The processing yield for portions with high economical value in catfish is presented in Table 3.

**Table 3.** Processing yield for portions with high economical value at catfish.

| Cut Part | RG (n = 50) | | | AG (n = 50) | | | *p*-Value |
|---|---|---|---|---|---|---|---|
| | Mean ± SEM | Min. | Max. | Mean ± SEM | Min. | Max. | |
| Live mass (g) | 1784.91 ± 37.43 | 1252.59 | 2192.54 | 1840.71 ± 30.25 | 1386.04 | 2152.35 | 0.0793 |
| Carcass mass (g) | 1598.39 ± 21.97 | 1156.50 | 1928.69 | 1659.40 ± 27.93 | 1260.44 | 1940.07 | 0.3781 |
| Carcass yield (%) | 89.55 ± 0.71 | 86.89 | 91.68 | 90.15 ± 1.13 | 86.81 | 92.89 | 0.1028 |
| Torso mass (g) | 1159.48 ± 17.99 | 805.13 | 1385.39 | 1124.31 ± 25.31 | 835.72 | 1324.75 | 0.0083 |
| Torso yield (%) | 64.96 ± 0.81 | 61.70 | 67.24 | 61.08 ± 0.81 | 58.79 | 63.07 | 0.3973 |
| Fillet mass (g) | 825.17 ± 13.44 | 610.74 | 948.33 | 830.35 ± 14.99 | 621.89 | 960.41 | 0.1629 |
| Fillet yield (%) | 46.23 ± 0.50 | 44.12 | 48.80 | 45.11 ± 0.55 | 43.10 | 47.12 | 0.0793 |

SEM: Standard error of mean. Analysis of variance: check *p* values per row.

The carcass yield revealed 0.67% better values in the AG fish compared to the RG fish while the catfish from the river had better efficacy for torso and fillet yields by 6.35% and 2.48%, respectively.

*3.3. Qualitative Flesh Production*

A comparative evaluation of losses and water content from European catfish refrigerated flesh in different storage periods is presented in Table 4.

**Table 4.** Comparative evaluation of losses and water content in European catfish refrigerated fillets throughout different storage periods.

| Storage Interval (Days) | n | Group | Losses (%) | Water (%) |
|---|---|---|---|---|
| | | | Mean ± SEM | Mean ± SEM |
| 0 | 6 | AG | 100 ± 0.00 | 77.80 ± 1.00 |
| | 6 | RG | 100 ± 0.00 | 78.19 ± 2.02 |
| *p*-value | | | - | 0.3039 |
| 3 | 6 | AG | 97.61 ± 0.98 | 76.60 ± 1.23 |
| | 6 | RG | 98.12 ± 1.67 | 77.26 ± 1.40 |
| *p*-value | | | 0.1872 | 0.1762 |
| 6 | 6 | AG | 93.56 ± 2.85 | 74.26 ± 2.64 |
| | 6 | RG | 94.58 ± 1.34 | 74.88 ± 1.12 |
| *p*-value | | | 0.3633 | 0.1906 |

**Table 4.** *Cont.*

| Storage Interval (Days) | n | Group | Losses (%) Mean ± SEM | Water (%) Mean ± SEM |
|---|---|---|---|---|
| 9 | 6 | AG | 90.48 ± 2.72 | 72.74 ± 2.25 |
|   | 6 | RG | 89.87 ± 3.25 | 72.63 ± 3.21 |
| *p*-value |   |   | 0.1964 | 0.3692 |
| 12 | 6 | AG | 88.79 ± 1.95 | 72.18 ± 2.07 |
|   | 6 | RG | 89.05 ± 2.72 | 72.18 ± 2.61 |
| *p*-value |   |   | 0.1408 | >0.9999 |
| 15 | 6 | AG | 87.12 ± 2.52 | 70.90 ± 2.17 |
|   | 6 | RG | 87.87 ± 2.03 | 71.47 ± 2.42 |
| *p*-value |   |   | 0.2386 | 2.2861 |

SEM: Standard error of mean. Analysis of variance: check *p* values per column, for each storage period.

Certain amounts of water and nutrients were lost from the flesh throughout the experimental period. Water loss reached 8.87% in the AG samples and 8.59% in the RG ones.

The evaluation of dry matter constituents from the European catfish in relation to the lastingness of the storage period is presented in Table 5.

**Table 5.** Evaluation of dry matter constituents from European catfish flesh in different storage periods.

| Storage Period (Days) | n | Group | Ash (%) Mean ± SEM | Proteins (%) Mean ± SEM | Lipids (%) Mean ± SEM |
|---|---|---|---|---|---|
| 0 | 6 | AG | 1.07 ± 0.00 | 17.75 ± 1.08 | 3.38 ± 0.22 |
|   | 6 | RG | 1.11 ± 0.04 | 18.08 ± 1.27 | 2.62 ± 0.16 |
| *p*-value |   |   | 0.0782 | 0.7320 | 0.0373 |
| 3 | 6 | AG | 1.05 ± 0.02 | 16.65 ± 0.82 | 3.31 ± 0.18 |
|   | 6 | RG | 1.08 ± 0.04 | 17.01 ± 0.89 | 2.77 ± 0.13 |
| *p*-value |   |   | 0.0836 | 0.6592 | 0.0459 |
| 6 | 6 | AG | 1.05 ± 0.05 | 15.21 ± 0.75 | 3.04 ± 0.16 |
|   | 6 | RG | 1.07 ± 0.04 | 16.22 ± 0.92 | 2.42 ± 0.12 |
| *p*-value |   |   | 0.0919 | 0.3182 | 0.0428 |
| 9 | 6 | AG | 1.03 ± 0.08 | 13.85 ± 0.72 | 2.87 ± 0.23 |
|   | 6 | RG | 1.04 ± 0.05 | 14.05 ± 0.76 | 2.15 ± 0.14 |
| *p*-value |   |   | 0.1305 | 0.7855 | 0.0285 |
| 12 | 6 | AG | 1.02 ± 0.07 | 13.04 ± 0.71 | 2.56 ± 0.24 |
|   | 6 | RG | 1.03 ± 0.09 | 13.85 ± 0.92 | 1.98 ± 0.12 |
| *p*-value |   |   | 0.1287 | 0.4348 | 0.0319 |
| 15 | 6 | AG | 1.00 ± 0.10 | 12.84 ± 0.56 | 2.38 ± 0.13 |
|   | 6 | RG | 1.01 ± 0.07 | 13.53 ± 1.02 | 1.86 ± 0.10 |
| *p*-value |   |   | 0.1149 | 0.4817 | 0.0402 |

SEM: Standard error of mean. Analysis of variance: check *p* values per column, for each storage period.

The protein levels decreased by 27.66% in the AG samples and 25.16% in the RG samples while the loss of lipids was 29.58% in AG flesh and 29% in RG flesh. Total minerals (crude ash) decreased by 6.54% in AG samples and 9% in RG flesh by the end of the 15 days of storage.

### 3.4. Fatty Acids Profile and Sanogenic Indices

The profile of fatty acids from the European catfish flesh and sanogenic indices evaluation is presented in Table 6. Analysis was carried out on fillets obtained from the catfish with the age of two summers issued from the farm (AG) or the wild environment (Prut River) (RG).

**Table 6.** Fatty acids profile (g/100 g total FAME) and sanogenic indices of European catfish fillets.

| Fatty Acids | AG (n = 6) | RG (n = 6) | *p*-Value |
| --- | --- | --- | --- |
| | Mean ± SEM | Mean ± SEM | |
| C 14:0 | 3.08 ± 0.078 | 1.54 ± 0.054 | $4.95 \times 10^{-9}$ |
| C 14:1 | ND | 0.32 ± 0.012 | - |
| C 16:0 | 11.54 ± 0.113 | 15.16 ± 0.399 | $1.46 \times 10^{-5}$ |
| C 16:1 | 3.66 ± 0.069 | 8.39 ± 0.730 | $1.20 \times 10^{-8}$ |
| C 18:0 | 2.92 ± 0.080 | 3.69 ± 0.196 | $5.20 \times 10^{-5}$ |
| C 18:1 n−9 | 19.98 ± 0.397 | 24.83 ± 1.492 | $9.53 \times 10^{-5}$ |
| C 18:2 n−6 | 8.25 ± 0.202 | 4.79 ± 0.397 | $3.75 \times 10^{-8}$ |
| C 18:3 n−3 | 1.70 ± 0.021 | 3.83 ± 0.218 | $1.41 \times 10^{-9}$ |
| C 20:0 | 0.18 ± 0.005 | 0.19 ± 0.017 | 0.1478 |
| C 20:1 n−9 | 5.75 ± 0.195 | 1.82 ± 0.100 | $1.23 \times 10^{-10}$ |
| C 20:2 n−6 | 0.44 ± 0.006 | 0.49 ± 0.010 | 0.0109 |
| C 20:4 n−6 | 0.25 ± 0.008 | 2.12 ± 0.042 | $6.56 \times 10^{-12}$ |
| C 20:3 n−3 | 5.51 ± 0.198 | 0.27 ± 0.007 | $2.93 \times 10^{-13}$ |
| C 20:5 n−3 | 3.57 ± 0.083 | 3.09 ± 0.125 | 0.0001 |
| C 22:0 | 0.49 ± 0.004 | 0.80 ± 0.010 | $9.14 \times 10^{-9}$ |
| C 22:5 n−6 | 0.17 ± 0.001 | ND | - |
| C 22:5 n−3 | 2.09 ± 0.014 | 1.91 ± 0.059 | 0.0293 |
| C 22:6 n−3 | 7.17 ± 0.187 | 8.61 ± 1.173 | 0.0003 |
| Σ SFA | 18.21 | 21.37 | |
| Σ MUFA | 29.38 | 35.36 | |
| Σ PUFA | 25.58 | 25.11 | |
| n−3 | 20.03 | 17.71 | |
| n−6 | 9.12 | 7.39 | |
| n−3/n−6 | 2.20 | 2.40 | |
| n−6/n−3 | 0.46 | 0.42 | |
| PUFA/SFA | 1.40 | 1.17 | |
| USFA/SFA | 3.02 | 2.83 | |
| PI | 11.65 | 12.45 | |
| AI | 0.41 | 0.35 | |
| TI | 0.22 | 0.27 | |
| HFA | 14.62 | 16.70 | |
| hFA | 45.56 | 49.94 | |
| h/H | 3.12 | 2.99 | |

SEM: Standard error of mean. Analysis of variance: check *p* values per row. PI: polyunsaturated index, TI: thrombogenic index, AI: atherogenic index, HFA: hypercholesterolemic fatty acids (C12:0 + C14:0 + C16:0), hFA: hypocholesterolemic fatty acids (C18:1 + polyunsaturated FA), h/H: hypocholesterolemic/hypercholesterolemic FA, ND: not detectable.

Out of the five PUFAs found in the studied European catfish, the most occurring in the AG samples were linoleic acid (8.25 g/100 g total FAME), eicosatrienoic acid (5.51 g/100 g total FAME), and eicopentaenoic acid (3.57 g/100 g total FAME). In RG fish, the PUFA with the highest proportion was linoleic acid as well (4.79 g/100 g total FAME) followed by alfa linolenic acid (3.83 g/100 g total FAME) and eicosapentaenoic acid as well (3.09 g/100 g total FAME).

The sum of the fatty acids was 17.35% higher in the RG samples than the AG samples for SFA and 20.35% higher for MUFA while lower for PUFA by 1.84% in river-originating fish compared to aquacultured ones.

Significant amounts of polyunsaturated fatty acids were measured in European catfish flesh with levels above 25 g/100 g total FAME, suggesting a high quality of inner lipids, a fact also highlighted by the good values of the sanogenic indices in both fish groups: PI = 11.65, TI = 0.22, and AI = 0.41 in AG samples and PI = 12.45, TI = 0.27, and AI = 0.35 in RG flesh.

## 4. Discussions

### 4.1. Morphometry and Body Indices

Dimensional and gravimetric investigations were carried out on 50 individuals/group that were aged two summers with body mass values close to the means of the groups.

No statistically significant differences occurred for body mass between groups. Differences can be attributed to environmental conditions because the vastness of the Prut hydrographic basin, permanent water stream, and lower feed quantity versus the environment provided by the farm influence, as was demonstrated, a growth rhythm. The obtained values fell within the literature limits [55,56].

Total length (L) values were statistically different ($p < 0.01$) between groups.

Standard length (l) represents a very important morphometric assessment because it highlights the anatomical portion that presents a direct interest for consumers [57]. Between groups, it presented close values of $58.74 \pm 0.46$ cm (AG) and $60.04 \pm 0.40$ cm (RG) ($p > 0.05$).

Head length (lh) is an important biometric assessment in fish because this cut does not have a high demand in consumption. So, in artificial selection, individuals with a lower HL related to total body length are preferred [58]. For the AG group, the HLs reached $11.50 \pm 0.11$ cm (18.12% from L) while in the RG group, they reached $12.48 \pm 0.12$ cm (18.21% from L), with significant differences between groups ($p < 0.01$) (Table 1).

Maximal body heights (H) presented significant statistical differences between groups ($p < 0.01$) (9.5 to 11.8 cm in AG, 8.4 to 11.3 cm in RG).

Maximal body circumference (C) is run on the anatomical portion in which the body has the highest thickness and respective height [53]. No significant differences occurred between groups. In the AG group, C reached 31.29 cm and was 2.76% less than in the RG group.

Maximal body thickness (T) was maximal in the RG group at $7.08 \pm 0.09$ cm, respectively, 1.97% higher than the one in the AG group.

Profile index (PI) highlights the body format and represents a rate between the body standard length (cm) and its maximal height (cm). Low values for this index suggest a convex aspect of the dorsal line, which, in practice, indicates the muscle mass is well represented at the dorsal level [59]. Highly significant differences between means occurred between groups ($p < 0.001$), with a lower value in aquaculture fish, proving the influence of artificial selection of the European catfish population at the farm level.

The Fulton coefficient (FC) indicated a very good maintenance state, acquired by the remarkable adaptability of the studied individuals to environmental conditions as well as through an optimal nutrient uptake from the water supply [33,60]. This index ranged between $0.91 \pm 0.01$ (AG) and $0.82 \pm 0.01$ (RG) ($p < 0.05$).

The quality index (QI) suggested a rich muscular mass. Considering that European catfish have quite a serpent-shaped body, the values of those indexes as well as the other

indices must not be compared with the values obtained for other fish species [61]. The QI reached $1.88 \pm 0.01$ in the AG group and $1.87 \pm 0.02$ in the RG group ($p > 0.05$), proving the quality of biological material in both groups.

The thickness index (TI) expresses the muscle width from the backbone region in relation to the body's maximal height, proving information regarding the fattening state or even about the fish's body format [21]. The means for this index were $64.29 \pm 0.61$ (AG) and $70.77 \pm 1.04$ (RG). Associating those values with the ones of the profile index, we can conclude that differences regarding body format exist between these two populations because at similar values of body mass and standard body length, the total maximal height for fishes from the Prut River was significantly reduced without a negative impact on the thickness of back musculature. This aspect can be interpreted as an adaptation to environmental conditions because a dorso-ventral body flattening opposes less resistance to a constant water flow.

The fleshy index (FI) represents the rate between head length and body standard length. This index has great importance because in production, the goal is to raise individuals with an optimal body format with economically relevant anatomical regions in greater proportion. The lower the FI, the higher the torso proportion from the fish's standard length [62,63]. The FI in the AG group reached $19.58 \pm 0.14$, and the RG group was 5.82% higher, indicating a higher rate of head participation in relation to fish standard length in river fish. These index values can also be considered as adaptations of individuals from the Prut River to environmental conditions, considering that feeding in the wild is not as facile as at the farm; therefore, a larger oral cavity can facilitate prey catching.

The achieved profile index values suggest a higher back and much more voluminous dorsal musculature in farmed catfish while the thickness index shows that catfish from the natural environment have a thicker body as an adaptation to environmental conditions. Higher values of the Fulton coefficient for farmed fish reveal their better development, and the lower values of the quality index (Kiselev) depict a richer muscular mass in farm vs. river-originating fish. The lower values of the fleshy index for the farm fish indicate better fleshiness in comparison with catfish from the natural environment. Considering mind–body indices and coefficients for both categories of fish, it can be stated that the aquaculture significantly and positively impacted body development in comparison with the natural environment.

*4.2. Quantitative Flesh Production*

Individuals captured from the Prut River had yields of 89.55% for carcasses, 64.96% for torsos, and 6.23% for fillets while the farmed fish achieved a 90.15% carcass yield, 61.08% torso yield, and 45.11% fillet yield. The obtained means were significantly different only for torso yields ($p < 0.05$). No major differences regarding the slaughtering yield occurred for the catfish originating from both environments. The obtained data fell within the limits in the literature for this species [8,19,64].

In a similar study, Jankovka et al. (2006) [19] analysed catfishes with different origins and mentioned close values between the yields of fish from a natural environment (carcass 90.75%; torso 60.08%; and fillets 42.79%) and those that were farmed (carcass 90.76%; torso 60.86%; and fillets 45.11%).

*4.3. Qualitative Flesh Production*

Due to its proximate composition, fish flesh is placed among the products with high biological value, mainly due to the protein quality which contains almost all the essential amino acids. The high quality of the fish flesh is supported by the reduced quantity of the connective tissue in muscles. Hussain et al. (2011) [65] assigned a digestibility coefficient for fish flesh of around 97% based specifically on the reduced quantity of the connective tissue.

Lipids from fish flesh have high amounts of unsaturated fatty acids, which are beneficial to consumers' health but provide a negative influence on the stability of muscle tissue, favouring its rapid alteration mostly through oxidation [66,67].

In skinless fillets, the mean value for relative moisture was 77.80% in AG samples and 78.19% in RG samples.

During storage periods, the mean values of water between the groups did not exceed a percentage difference higher than 0.66% (RG vs. AG). No statistically significant differences occurred between groups for the water content in the flesh; therefore, the rearing environment did not have an influential role in defining flesh proximate composition. A much more important role in the chemical composition of the flesh is attributed to the applied preservation method and storage period [60]. In a 15-day period, European catfish fillets refrigerated in chilled air flow recorded mass losses of 12.88% and 12.13% for the AG group and RG group, respectively, versus the initial rate. In the AG fish, the losses meant a decrease of 6.90 percentage points of moisture versus the first evaluation. In the RG samples, the water content decreased by 6.72 percentage points throughout the same storage period.

The literature provides much information on the water content of European catfish flesh [68,69]. In our study, flash moisture ranged between 77.68 ± 0.45% and 79.45 ± 0.37% in relation to feeding type. Close values for the same trait were reported by other authors [15,16,55].

Flesh protein quantity in the RG group had a decreasing trend vs. the AG group due, most probably, to the samples' more intense hydrolysis. This flesh denaturation favours the development of alternative bacteria. So, AG samples indicated a protein mean value of 17.75% compared to 18.08% in the RG group. At the end of the 15 storage days, the protein level reached 12.84% in the AG group and 13.53% in the RG group. During the six evaluation sessions for this constituent, no statistical significance occurred between groups. In a study on the chemical composition of flesh gathered from aquaculture featuring catfishes with live weights close to the aquaculture catfishes analysed in the current study (1813.51 g), Honzlova et al. (2021) [16] mentioned comparable values for protein content (16.35–18.12%).

Generally, in most fish species, there is a correlation between lipids content and the conditions provided in different rearing systems; this situation is also valid for European catfish [20,70,71]. In the present study, total lipids values greater by 22.5% occurred in aquaculture catfishes vs. the river-originating ones, thus highlighting the influence of feeding with mixed fodders on the flesh fattiness. Concretely, the recorded limits of lipids varied during the 15 storage days between 3.38% and 2.38% for AG fishes and 2.62–1.86% for RG catfishes. There were also observed fluctuations due to the activity of alteration microorganisms as well as oxidative processes that appeared on the sample surface [64,69,72]. Moreover, for the case of lipids, the literature indicates close values to those obtained in the current study [15,16]. Linhartova et al. (2018) [73] indicated catfish flesh lipids contents of 4.13% in intensive rearing system samples and 2.97% in the semi-intensive system.

Statistically, during the whole storage period, the differences with statistical significance ($p < 0.01$) can be accounted for by the influence of the rearing environments. So, due to water action as well as due to lower food quantity, related to water volume unit, fish from the wild environment are prone to exhibit higher effort in comparison with the ones from the farm where, due to a very high density of fish–prey, the latter have a much more sedentary lifestyle. Regarding the dynamics of these three traits, a descending trend was noticed, indirectly proportional to the prolongation of the storage period.

Ash quantity in AG samples decreased by 0.07 percentage points compared to the initial quantity while RG samples decreased by 0.10 percentage points.

### 4.4. Fatty Acids Profile and Sanogenic Indices

In accordance with the data presented in Table 6, out of the 18 identified fatty acids, oleic acid was most present (19.98–24.83 g/100 g total fatty acids) followed by palmitic acid (11.54–15.16 g/100 g total fatty acids), a fact also outlined by other authors [73].

Statistically, significant differences occurred between the aquaculture and captured fish for most of the analysed fatty acids ($p < 0.01$) except for arachidonic and eicosadienoic acid.

In lipids constitution, the highest proportion was occupied by MUFA (29.38–35.36%) followed by PUFA (25.11–25.58%) and SFA (18.21–21.37%), indicating a high quality of fats in the analysed European catfish fillets. Higher MUFA and SFA proportions were obtained in the wild samples versus the farmed fish which presented higher values for PUFA.

Close values were reported by Linharthova et al. (2018) [73] for catfishes intensively and semi-intensively reared: MUFA between 37.36 to 41.61%, PUFA 28.86 to 34.61%, and SFA 22.25 to 24.23%. High contents of MUFA and PUFA, known for their beneficial effects on human health, especially as protective against cardiovascular diseases [74,75], make the European catfish flesh an important source of "good fats". The assimilation degree of fish fat is better in human consumers versus other dietary fats due to the higher presence of linoleic, linolenic, arachidonic, eicosapentaenoic, docosapentaenoic, and docosahexaenoic acids. The high rate of PUFA n−3/n−6 in fillets obtained from *Silurus glanis* reared in an intensive rearing system and extensive system could also have a protective effect against breast cancer [76].

The high level of the polyunsaturated index (PI) in fillets from aquaculture and captured European catfish (11.65–12.45) outlines a high PUFA level, relevant for human health due to implications for the adjustment of cholesterol levels in blood [77,78]. Bearing in mind the fact that the PI for the RG samples was 6.9% higher, this places the fishes from the natural environment on a better rank compared to the ones from aquaculture.

From a human health perspective, the thrombogenic index (TI) and atherogenic index (AI) highlight the predisposition for cardiovascular disease occurrence in consumers and express the relation between saturated (prothrombo/atherogene) and unsaturated lipids (antithromobo/atherogene) [48,79]. The AI value calculated for *Silurus glanis* samples in AG (AI = 0.41) was 39.02% lower compared to carp (0.57) and 58.53% less than in trout (0.65). In samples from captured *Silurus glanis*, the AI value of 0.35 was also lower than in carp (by 62.85%) and in trout (by 85.71%) [80]. The highest values for the AI were reported by Kucukgulmez et al. (2018) [81] for two fish species from salty waters (AI = 1.22). The TI calculated for the studied catfish was lower in comparison with other freshwater fish species (carp TI = 0.63, trout TI = 0.49, and paddlefish TI = 0.39) [80,82], suggesting a possible lower tendency in consumers for blood clots formation. Comparatively analysing the data obtained in the current study, we observe the fact that fish from the RG had an 14.6% better AI in comparison with AG fish. Regarding the thrombogenic index, we observed that aquaculture fishes are superior to the ones from the natural environment by around 22.7%.

European catfish fillets were characterised by a quite high occurrence of fatty acids with hypocholesterolemic effects (hFA) (45.56–49.94), aspects resulting from the high rate of the h/H FA (2.99–3.12). The h/H index suggests the presence of enough valuable lipids with the potential to decrease consumers' blood serum cholesterol [51]. It was observed that catfish from the natural environment could have a better hypocolesterolemic effect (hFA higher at RG with 9.61% and h/H lower with 4.16%) than fish from aquaculture.

## 5. Conclusions

A morphometric assessment revealed that catfish from a natural environment had close values to the ones from the farm with a slight superiority of aquaculture specimens.

In terms of chemical composition, fish origin has particularly influenced the lipids content, found in higher amounts in farm fishes, due to environmental conditions as well as to the facile feed availability.

Flesh storage in a refrigerated state for 15 days leads to chemical modifications by losing tissue water, especially throughout the first storage phases, followed by a decrease of nutrients (proteins, fats) due to processes associated with the exudation and degradation of the flesh.

The fatty acids profile and sanogenic indices suggested the better quality of catfish versus other freshwater commonly consumed fish species due to the significant proportion of PUFA and better sanogenic indices. A better lipids profile and sanogenic indices values occurred in wild catfish versus farmed ones.

As a research follow-up, it would be suitable to have an analysis of the pollution degree of catfish environments and the possible transfer of such pollutants through the food chain of water–fish–human consumer.

**Author Contributions:** Conceptualization, C.S. and E.M.; methodology, B.P.; software, E.M. and B.-V.A.; validation, B.P. and D.S.; formal analysis, C.S. and E.M.; investigation, C.S., E.M., B.P. and B.-V.A.; data curation, E.M. and D.S.; writing—original draft preparation, C.S. and E.M.; writing—review and editing, C.S., B.-V.A. and D.S.; supervision, C.S. All authors have read and agreed to the published version of the manuscript.

**Funding:** This research received no external funding.

**Institutional Review Board Statement:** Not applicable. No animals were used for applying experimental factors on them or to measure their effect through reasoning criteria. The fish individuals used for morphometry and flesh sampling were issued from a fishery or public river and were part of a group marketed afterwards.

**Informed Consent Statement:** Not applicable.

**Conflicts of Interest:** The authors declare no conflict of interest.

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
