# Peer review of "Quantitative and Qualitative Assessment of European Catfish (Silurus glanis) Flesh"

_agriculture, doi:10.3390/agriculture12122144_

Round 1

Reviewer 1 Report

The work submitted for review on the evaluation of carcass and meat quality from two maintenance environments of catfish species. The study used a small number of fish from each group (n = 10). In such studies, it should be much larger. This fact is confirmed by the obtained differences between the groups, or actually their lack. Probably the authors assumed that due to the large number of calculated indicators for carcasses and fatty acids, it will make their publication appreciated.

I believe that the research topic does not contribute much to science. Little is known about the water quality in these two environments, and this determines the results obtained in carcass quality studies. Moreover, we know nothing about their fish diet, which has a particular impact on the results obtained in the field of fatty acids. There is no detailed information on the storage of fish meat - sample size, storage conditions, etc. The confirmation of what I wrote are the general conclusions that indicate that it is worth eating catfish meat because it contains a lot of valuable fatty acids. I do not think that the conclusion is that the fish from the farm obtained better meat production results than those from the natural environment (river).

I believe that this work should be published in a journal promoting the consumption of catfish, after it is supplemented with results on the content of elements in the meat, including toxic ones, which are the result of river pollution.

Author Response

Honourable reviewer,

down bellow we reply punctually to your questions.

  1. n = 10

In according with your suggestion we reanalysed the data base, generating new values for Tables 1, 2 and 3 (n = 50).

2. Data regarding water quality and feeding

In accordance with your suggestion we introduce at 2. Materials and methods - 2.2. Physical-chemical parameters of water and 2.3. Catfish feeding

 2.2. Physical-chemical parameters of water

Water temperature (°C), had close values for both rearing systems. In the studied period (March-October), water from system La recorded in March a temperature of 10.2°C, and for system Lc, water temperature was 9.2°C. The highest thermal values were recorded in July (25.5°C) for fish farm, respectively in August (24.6°C) for Prut River. If at the beginning of the studied period (March-August) was observed a slightly difference regarding temperature, with a little bit higher values in favour of fish farm, starting with September were observed values slightly higher for the water from Prut River. Water pH varied between 7.2 and 7.9 for the water from fish farm, while for the Prut River water, pH oscillated between 7.1 and 7.5, the obtained values being into the normal parameters for a good development of studied species, in according with Order MAPM (Ministry of Water and Environment from Romania) 1.146 [23]. Quantity of dissolved oxygen was between 4.09 mg/l and 8.85 mg/l for fish farm water, while in natural environment, dissolved oxygen varied from 8.06 mg/l to 10.12 mg/l during analysed period (March-October). These values are into the necessary limits for a normal development of Silurus glanis species. The other water’s physical-chemical parameters analysed in the current study were chlorides (Cl-) which recorded values between 60.21 – 105.91 mg/l for fish farm and between 74.26 – 101.34 mg/l in Prut River; nitrites (NO2) with values between 0.02 - 0.15 mgN/l at fish farm and between 0.08 – 0.21 mg/l in Prut River; nitrates (NO3 ) with values from 0 up to 2.51 mgN/l at fish farm and between 1.12 – 2.14 mgN/l in Prut River; ammonium (NH4 +) with values from 0.03 till 0.14 mgN/l at fish farm and between 0.14 – 0.35 mgN/l in Prut River and phosphates (PO43) with values 0 till 0.12 mgP/l at fish farm and between 0.10 – 0.21 mgP/l in Prut River. In concordance with the obtained values we can affirm that the water from the both rearing systems was suitable for a normal development of fishes, being placed in 2nd and 3rd quality category – specific for fish farming systems [23].

Rearing of European catfish at the studied fish farm was realised in 2 ponds, each of them with 30 hectares, in polyculture with carp (Ciprinus carpio) 82%, silver carp (Hypophthalmichtys molitrix) and bighead carp (Aristichtys nobilis) 12%, with one year older than European catfishes and corporal masses of around 500 g. Rate of European catfish in both ponds was 6% (750 individuals), when populating the ponds catfishes had masses of around 250 g and at the end of growing period were recorded corporal masses which varied between 1.3 kg and 2.2 kg.

 2.3. Catfish feeding

Feeding of catfishes from aquaculture was realised with mixed fodders, the utilised quantity varied between 120-180 kg/day function of month, in 6 portions/day each having 20-30 kg/portion. Mixed fodder was purchased from the specialised stores, and had the following physical-chemical characteristics: granule dimension – 6 mm, dry matter (DM) 89%, crude protein (CP) 54%, crude fat (CF) 20%, crude ash 9%, crude cellulose (CC) 1%, P 1.1%, vitamin A 15000 IU/kg, vitamin D3 1800 IU/kg, vitamin E 105 mg/kg, vitamin C 280 mg/kg. Energetic value was 20.6 MJ/kg DE.

Catfishes from Prut River were natural feed with earthworms, snails, insects, tadpoles, frogs and fishes which are populating this river, such as: carp (Ciprinus carpio), Gibel carp (Carassius auratus gibelio), common bream (Abramis brama), common bleak (Alburnus alburnus), chub (Leuciscus cephalus), common rudd (Scardinius erythrophthalmus) and common dace (Leuciscus leuciscus).

3 Flesh storage – samples’ dimension and storage conditions

In according with your suggestion we introduce at 2. Materials and methods - 2.6. Sampling

2.6. Sampling

In the current was applied a refrigeration of fishes just after fishing, followed by slaughtering and obtaining of skinless fillets. Samples had a mass of 100 g and were individually refrigerated (Figure 1) at a temperature between 2°C and 4°C, and an air relative moisture of 80 – 85%.

4. Conclusions

In concordance with your suggestion we reformulated conclusion 1.

Morphometric evaluation of studied fishes revealed the fact that catfishes from natural environment presented close values to the ones obtained for farm catfishes, with a slightly superiority for fishes reared in aquaculture. This fact of state was confirmed by the obtained values for the majority of analysed quantitative parameters.

  1. Addition with toxic substances – river pollution

In concordance with your suggestion regarding pollutants’ analysis, we aim that in the future to analyse those aspects.

At – Conclusions

As a further research, will be suitable to have an analysis connected with pollution degree of European catfishes’ life environments and the possible presence of pollutants in their flesh.

Reviewer 2 Report

These days consumers are very conscious about food safety and quality. Thus it is always interesting to know about quality aspects of wild caught and farmed fish. However the MS in present needs lots of improvement. Few of the suggestion are given below.

1. Change in title like “Assessment of the Quantitative and qualitative assessment of European catfish (Silurus glanis) flesh

2. The introduction should be more concise with clear objective of the study and practical utility 

3. Material and method description seems to too long. It may be more concise.  

4. Whether there was significant difference in quality of flesh eg fatty acid profile or not ? Test of significance may be incorporated and reason for the difference may be provided and discussed with relevant literature.

5. Conclusion may be made lot more clarity with result data obtained.

Author Response

Honourable reviewer,

in accordance with your suggestions we made the following changes as are presented below.

  1. Title change

In according with your suggestion we change the title of the article:

Quantitative and qualitative assessment of the European catfish (Silurus glanis) flesh
  1. More concise introduction with clear aims and practical utility

In according with your suggestion we reformulated the introduction:

In the current paper we aimed the following goals:

  • obtain of quantitative information by measuring and weighting of certain anatomical parts for European catfish reared in aquaculture, as well as from natural environmental and through calculation of certain rations between dimensions which provide information regarding productive potential as well as an appreciation of their maintenance state and their adaptability to the assured environment conditions;
  • tracking the modifications related to flesh chemical composition of European catfish reared in aquaculture and from natural environment, during storage in refrigeration state for 15 days as well as the establishment of fatty acids profile and calculation of sanogenic indexes for the analysed flesh.

Practical utility of the study is given by the knowledge and understanding of qualitative and quantitative characteristics of Silurus glanis species flesh obtained in different rearing environments and preserved by refrigeration. So, we could show that for European catfish flesh gathered from aquaculture and from capture, kept at refrigeration temperature, could be optimal for consumption up to 9 days.

  1. Material and method - more concise

In according with your suggestion we modify at 2. Materials and methods – 2.8 Analysis of fatty acids profile and nutritional quality of lipids

  1. Statistical differences at fatty acids

    In according with your suggestion, we modified table 6 and we introduced at 4.4. Fatty acids profile and sanogenic indexes of European catfish meat (Silurus glanis):

Statistically speaking, were observed significant differences between aquaculture catfishes and the captured ones for the majority analysed fatty acids, with the exception of Arachidonic acid and Eicosadienoic acid.

  1. More clear conclusions

In concordance with your suggestions we modified the conclusions:

Morphometric evaluation of studied fishes revealed the fact that catfishes from natural environment presented close values to the ones obtained for farm catfishes, with a slightly superiority for fishes reared in aquaculture. This fact of state was confirmed by the obtained values for the majority of analysed quantitative parameters.

Chemically speaking, provenance source influence especially flesh fat content, which was higher at farm fishes, due to environmental conditions, as well as to feed availability.

Flesh storage in refrigerated state for 15 days, leads to chemical modifications by: losing of tissue water especially in the first storage phases, followed by a decreasing of nutritive elements quantity (proteins, fats) due to processes associated to degradation of fish flesh.

Reviewer 3 Report

The study is interesting and author is try to compare flesh quality between farmed and wild european catfish. However there are few issues need to be solved before this paper can be published. The issues are as follows:

Title: Accepted

Abstract: well elaborated

Introduction:

Line 54, Line 58-59: please refer to latest reference for total production of the fish. Perhaps in 2021?

Materials and methods

Please add references in section 2.6

Results: Please elaborate more the outcomes of the study for example which treatment is highest or lowest with or without significant

Discussion

Author need to discuss in detail each finding of the study by compared to other study or elaborate what factors that contribute to the outcomes

References

Many old references. Please make sure 75% of references are within 5 years source

Author Response

Distinguished reviewer,

in according with your suggestions we made the following changes.

  1. Introduction– recent citations

On line 54 and line 58-59, reference source 14 had the most recent FAO data regarding productions from pisciculture.

  1. Material and method

In according with your suggestion we introduce reference [52] at 2.9. Statistical analyses

  1. Results

In concordance with your suggestion we compare the obtained results, between fishes from natural environment and the ones from aquaculture:

For: 4.1. Corporal dimensions and corporal indexes

The obtained values for profile index suggest a higher back and a much more voluminous dorsal musculature at farm catfishes while thickness index show that catfishes from natural environment have a thicker body, as an adaptation to environment conditions. Higher values for Fulton coefficient, for farm fishes reveal a better development of those ones, and the lower values of quality index (Kiselev) suggest a rich muscular mass of those ones face to catfishes from natural environment. The lower values of fleshy index for farm fish indicate a better fleshy in comparison with the catfishes from natural environment. Bearing in mind corporal indexes and coefficients for both categories of analyzed fishes we could affirm that catfishes from aquaculture had a better development level face to the ones from natural environment.

For: 4.2. Quantitative meat production gathered from studied fishes

From the data obtained by us weren’t founded major differences regarding slaughtering yield for the catfishes reared in both rearing environments.

For: 4.3. Qualitative meat production gathered from studied fishes

Also from our study were revealed differences, higher values with 22.5% for lipid content at aquaculture catfishes face to the ones from environmental one, which highlighted the influence of feeding with mixed fodders for aquaculture fishes.

For: 4.4. Fatty acids profile and sanogenic indexes of European catfish meat (Silurus glanis)

From analysis of fatty acids resulted higher values for catfishes from natural environment, for MUFA and SFA while fishes from aquaculture recorded slightly higher values for PUFA.

Bearing in mind the fact that PI for Lc was with 6.9% higher place the fishes from natural environment on a superior rank face to the ones from aquaculture.

Analysing comparatively the data obtained in the current study, we observe the fact that fishes from Lc had an Atherogenic index better with 14.6% in comparison with the fishes from La. As regarding thrombogenic index, we observed that aquaculture fishes are superior to the ones from natural environment with around 22.7%.

From the differentially analysis of studied fishes, was observed that catfishes from natural environment could have a higher hypocolesterolemic effect (hFA higher at Lc with 9.61% and h/H lower with 4.16%) than fishes from aquaculture.

  1. Discussions

In according with your suggestion we compare the results obtained by us with other studies:

For: 4.2. Quantitative meat production gathered from studied fishes

Into a similar study, Jankovka et al., 2006 [19] analysing catfishes from two provenance environments, also mention, close values between fishes from natural environment (carcass 90.75%; trunk 60,08% and fillets 42.79%) and the ones from aquaculture (carcass 90.76%; trunk 60.86% and fillets 45.11%).

For: 4.3. Qualitative meat production gathered from studied fishes

In a study related to chemical composition of meat gathered from aquaculture catfishes with corporal masses close to aquaculture catfishes’ which were analysed in the current study (1813.51 g), Honzlova et al., 2021 [16] mentioned close values to the ones obtained by us regarding fillets’ protein content (16.35% – 18.12% protein).

Linhartova et al., 2018 [73] indicate values of 4.13% lipids in meat of catfishes reared into a intensive rearing system and 2.97% lipids for the catfishes which were reared into a semi-intensive system.

For: 4.4. Fatty acids profile and sanogenic indexes of European catfish meat (Silurus glanis)

Close values were reported by Linharthova et al., 2018 [73]. So, for catfishes which are intensive and semi-intensive reared, the above mentioned authors indicated for MUFA values between 37.36 and 41.61%, for PUFA 28.86 – 34.61% and for SFA 22.25 – 24.23%.

  1. References

In according with your suggestion we improved the reference list with articles published in the last 5 years.

Round 2

Reviewer 1 Report

I believe that the first sentence in Abstract should be removed because it does not add anything to the research results obtained. The Abstract should specify exactly what meat quality parameters were affected by the storage period. In my opinion, in order to increase the clarity of the tables, the p-value should be provided instead of the coefficient of variation, which is not analyzed in the discussion of the results.   I also believe that only the water or dry matter content should be left in table 4 as the water content is calculated from the determination of the dry matter in the meat.  

Author Response

Honourable Reviewer, 

In accordance with your suggestion we modified the abstract and we introduced a phrase for modifications of meat chemical parameters during storage period, as follows below:

”Keeping of catfish flesh in refrigeration conditions for 15 days led to chemical modifications. So, for AG water content decrease with 8.87%, proteins with 27.66% and the lipids with 29.58%. For RG the losses were 8.59% in water, 25.16% in proteins and 29%in lipids.”

We also modified the tables by introducing of p values and we eliminate the variation coefficient.  Also in Table 4 we eliminate the columns regarding dry matter.

Thank you very much once again!

Reviewer 2 Report

The changes made as per the earlier comment is satisfactory. However the extensive editing and improvement in English is required.

Author Response

Honourable Reviewer,

thank you for your suggestions and appreciations and we checked and improve the article.

Once again thank you very much.